# The Multifaceted Role of Regulatory T Cells in Sepsis: Mechanisms, Heterogeneity, and Pathogen-Tailored Therapies

**DOI:** 10.3390/ijms26157436

**Published:** 2025-08-01

**Authors:** Yingyu Qin, Jingli Zhang

**Affiliations:** Department of Pathogenic Biology and Immunology, Jiangsu Provincial Key Laboratory of Critical Care Medicine, School of Medicine, Southeast University, Nanjing 210096, China; jinglizhang_2004@163.com

**Keywords:** Tregs, sepsis immunopathology, functional heterogeneity, pathogen-tailored immunotherapy

## Abstract

Sepsis is a life-threatening condition caused by a dysregulated immune response to infection, characterized by an initial hyperinflammatory phase frequently followed by compensatory immunosuppression (CARS). Regulatory T cells (Tregs) play a critical, biphasic role: inadequate suppression during early hyperinflammation fails to control cytokine storms, while excessive/persistent activity in late-phase immunosuppression drives immune paralysis and secondary infection susceptibility. This review explores advances in targeting Treg immunoregulation across bacterial, viral, and fungal sepsis, where pathogenic type critically influenced the types of immunoresponses, shaping Treg heterogeneity in terms of phenotype, survival, and function. Understanding this multifaceted Treg biology offers novel therapeutic avenues, highlighting the need to decipher functional heterogeneity and develop precisely timed, pathogen-tailored immunomodulation to safely harness beneficial Treg roles while mitigating detrimental immunosuppression.

## 1. Introduction

Sepsis is a life-threatening systemic condition caused by a dysregulated host response to infection. It is primarily characterized by excessive systemic inflammation and immune system dysfunction, which can progress to multiple organ failure and is a leading cause of death [1]. Globally, sepsis affects over 48.9 million people annually, resulting in approximately 11 million deaths—accounting for roughly 20% of all global mortality [2].

Regulatory T cells (Tregs), defined by the expression of the transcription factor Forkhead box protein P3 (Foxp3), constitute a specialized subset of CD4^+^ T lymphocytes. They play a critical role in maintaining immune homeostasis and preventing autoimmunity through the suppression of excessive immune activation [3]. Dysregulation or functional impairment of Tregs contributes to the pathogenesis of diverse diseases, including autoimmune disorders, infections, and malignancies [4].

In sepsis, Tregs exert a dynamic and stage-dependent immunomodulatory effect during disease progression [5]. During the initial hyperinflammatory phase, the innate immune system rapidly detects pathogens, activating neutrophils, monocytes/macrophages, and dendritic cells. These cells release substantial amounts of proinflammatory cytokines (e.g., IL-6, TNF-α, IL-1β) to combat the infection. However, this intense inflammatory response triggers a compensatory anti-inflammatory response syndrome (CARS), during which Tregs undergo significant expansion and/or persistent activation [6]. While this response serves to limit excessive inflammation and protect host tissues from immunopathological damage, it simultaneously contributes to excessive immune suppression and dysfunction/exhaustion of effector T cells. This state of immunosuppression weakens host defense against secondary infections and increases mortality risk [7,8]. Furthermore, emerging evidence suggests that Tregs may also exhibit insufficient suppressive function during the initial hyperinflammatory phase, failing to adequately restrain the early proinflammatory cascade [9,10]. Therefore, the biphasic behavior of Tregs—characterized by inadequate suppression early in the hyperinflammatory phase and excessive suppression during the subsequent immunosuppressive phase—represents a key mechanism underlying immune dysregulation and adverse outcomes in sepsis. Due to the extensive regulatory role that Tregs play in the immune system, they represent a target with considerable therapeutic potential for sepsis. Therefore, this review aims to critically evaluate the pathogen-specific modulation of Treg heterogeneity and its functional consequences for immune dysregulation across the sepsis continuum. Crucially, we will emphasize how deciphering this pathogen-driven heterogeneity is pivotal for developing targeted, etiology-tailored immunomodulatory strategies.

## 2. Literature Search Methodology

Relevant publications (2015–2025) were systematically identified through PubMed and Scopus using combinatorial keywords: (“Treg” OR “regulatory T cell”) AND (“sepsis” OR “septic shock”) AND (“heterogeneity” OR “subtype” OR “plasticity”) AND (“bacterial” OR “viral” OR “fungal” OR “COVID-19”) AND (“immunosuppression” OR “immunoparalysis” OR “immunotherapy”). Articles underwent triage by title/abstract screening followed by full-text assessment for mechanistic and therapeutic relevance to pathogen-driven Treg dynamics in sepsis. Pre-2015 publications were also included to ensure a balanced perspective encompassing both foundational knowledge and recent advances.

## 3. Treg Heterogeneity

Depending on their origin, Tregs can be classified into thymus-derived regulatory T cells (tTregs) and peripherally induced regulatory T cells (pTregs). Among them, tTregs mature in the thymus, while pTregs originate from conventional CD4^+^Foxp3^−^ T cells and are induced to express Foxp3 and acquire immunosuppressive functions in peripheral tissues [11,12]. Additionally, iTregs refer to experimentally induced Tregs ex vivo, which exhibit significant immunosuppressive capabilities both in vitro and in vivo [13].

In terms of differentiation, mature Tregs have also been broadly categorized into two major subsets—resting/central Treg (cTreg, CD44^lo^CD62L^hi^, CCR7^+^) and effector/effector memory Treg (eTreg, CD44^hi^CD62L^lo^, CCR7^−^)—based on their differential expression of CD44, CCR7, and CD62L [14]. Compared with cTregs, eTregs exhibit enhanced functional adaptability and phenotypic diversity, characterized by the upregulation of specific transcription factors, chemokine receptors, and effector molecules, which enables effective suppression of immune responses within distinct microenvironments [4]. For example, during Th1-type inflammation, the transcription factor T-bet—indispensable for Th1 differentiation—is also induced in Tregs in both humans and mice and plays a pivotal role in suppressing Th1-mediated immune responses [5,10]. Similarly, interferon regulatory factor 4 (IRF4) expression in Tregs is essential across species for the effective inhibition of Th2-type immune responses [15,16]. Specific deletion of STAT3 in murine Tregs impairs their regulation of Th17-mediated immune responses, underscoring its non-redundant role in suppressing pathogenic inflammation (with analogous pathways conserved in humans) [17,18]. Paradoxically, under proinflammatory conditions (e.g., IL-6/IL-1β stimulation), STAT3 activation destabilizes Tregs by promoting Foxp3 downregulation and phenotypic plasticity. Moreover, although the transcription factor Bcl6 is a central regulator of follicular helper T cell (TFH) development, it is also expressed in Foxp3^+^ Tregs in both humans and mice, giving rise to a distinct subset known as follicular regulatory T cells (TFRs), which predominantly mediate immunosuppression during Th2-type immune responses [19]. GATA3 is a transcription factor driving type 2 immunity. GATA3 expression in Treg cells in both humans and mice is especially important for their tempering of Th2 responses in the intestine and the skin [20,21,22] Furthermore, Helios serves as a key signature marker of mouse and human tTregs, associated with a pronounced effector-like phenotype, enhanced suppressive capacity [23], and essential stability for Treg suppressive function [6,24]. 

The expression of certain surface molecules has also been shown to confer distinct functional properties of Tregs. For example, Tregs expressing the T cell immunoglobulin with ITIM domain (TIGIT) represent a highly suppressive subpopulation in both humans and mice that exhibits higher expression of suppressive molecules such as granzyme B and IL-10 [25]. As a co-inhibitory molecule, TIGIT exerts its immunosuppressive function through interaction with two ligands, CD155 (PVR) and CD112 (PVRL2), which are primarily expressed on antigen-presenting cells (APCs). These same ligands are also capable of binding to CD226 (DNAM-1) and CD96 receptors present on T and NK cells, thereby suppressing their activation [26]. Critically, TIGIT^+^ Tregs contribute to the selective Treg cell-mediated suppression of proinflammatory Th1 and Th17 cells but not Th2 responses by inducing the secretion of the soluble effector molecule fibrinogen-like protein 2 (Fgl2) [25]. Another key functional marker is CD39, the enzyme responsible for generating adenosine, a potent immunosuppressive agent. This enzyme is predominantly expressed by Tregs but not limited to Tregs. In mice, CD39 is found on nearly all CD4^+^ CD25^+^ T cells. However, in humans, its expression is limited to a specific subset of Foxp3^+^ regulatory effector/memory-like T (TREM) cells [27,28]. Importantly, CD39^+^ Tregs maintain superior functional stability under inflammatory stress and deploy potent adenosine-mediated suppression (via the ATP–CD39–CD73–adenosine axis and A_2_AR signaling) to effectively constrain effector T cell responses [29,30]. Taken together, immune microenvironments program Treg heterogeneity, conferring distinct suppressive functions via specialized transcription factors and surface molecules.

## 4. Immunoregulatory Network of Tregs in Sepsis

The complex immune response in sepsis creates a dynamically changing microenvironment that profoundly modulates the differentiation, functional activity, and phenotype of Tregs. This multifaceted regulatory process is driven by interconnected mechanisms such as cytokine signaling networks, direct intercellular interactions, and metabolic reprogramming (Figure 1).

### 4.1. Core Driving Role of Cytokines

Early in sepsis, levels of inhibitory cytokines such as TGF-β and IL-10 increase markedly alongside proinflammatory mediators (e.g., IL-1, IL-6, TNF-α), resulting in an “inflammatory–anti-inflammatory storm.” This early release of TGF-β and IL-10 serves as a compensatory mechanism to counteract excessive inflammation [31]. However, sustained elevation of these cytokines is predictive of immunosuppression, correlates strongly with adverse clinical outcomes, and further promotes the increase in Tregs [32,33].

As a critical inducer of Treg differentiation, TGF-β peaks during the acute phase of sepsis. It originates from a variety of cellular sources, including platelets, monocytes/macrophages, injured tissue epithelial cells (e.g., lung and kidney), myeloid-derived suppressor cells (MDSCs), and autocrine secretion by Tregs themselves, forming a multicellular collaborative network [34].

TGF-β1 exerts its core role by activating the Smad3 signaling pathway: the Smad2/Smad3 complex translocates into the nucleus, directly binding to conserved enhancer regions (e.g., CNS1) of the Foxp3 gene to initiate transcription [35]. Simultaneously, it recruits acetyltransferases such as Tip60/p300 to promote acetylation of Foxp3, thereby inhibiting its ubiquitin-mediated degradation [36]. Furthermore, a physical interaction between Smad3 and the Notch intracellular domain (NICD) synergistically enhances the activity of the Foxp3 promoter [37,38,39].

IL-10 is mainly secreted by M2 macrophages, MDSCs, natural killer (NK) cells, regulatory B cells (Breg), and Tregs themselves, enhancing Treg differentiation and function to form a positive feedback loop [40]. Upon receptor binding, IL-10 activates the JAK1/STAT3 signaling pathway, directly upregulating Foxp3 expression and promoting the differentiation of naïve T cells into Tregs [41]. Furthermore, IL-10 synergizes with TGF-β to augment Treg suppressive capacity and modulates metabolic reprogramming by enhancing Treg oxidative phosphorylation (OXPHOS), which supports their survival and functional stability [42].

IL-2, secreted by activated T cells, is also essential for Treg proliferation and survival [43,44]. Due to their lack of CD25 (IL-2Rα) expression and inability to secrete IL-2 themselves, immature Tregs are entirely dependent on paracrine signals. IL-2 enhances/stabilizes Foxp3 transcription via the JAK1/3-STAT5 pathway [45] and promotes Treg proliferation and survival through the PI3K-Akt-mTOR pathway [46].

IL-33, a member of the IL-1 family, is primarily produced by barrier cells such as endothelial cells, epithelial cells, and fibroblasts, which plays a dual role in sepsis [47]. Early on, it may inhibit TGF-β-mediated Treg differentiation, thereby aggravating inflammatory responses [48]. However, prolonged secretion contributes to sustained immunosuppression through modulation of other immune cell subsets, for instance, by promoting Group 2 innate lymphoid cell (ILC2)–M2 polarization, which in turn facilitates Treg proliferation and differentiation [49,50]. Notably, Tregs themselves express the IL-33 receptor ST2 and may expand in an ST2-dependent manner in response to IL-33 [51], suggesting the IL-33/ST2 axis directly participates in Treg-mediated immunosuppression.

### 4.2. Critical Impact of Metabolic Reprogramming

Severe metabolic disturbances induced by sepsis modulate the fate of Tregs via mechanisms involving nutrient deprivation, metabolite accumulation, and remodeling of energy-sensing pathways. Lactate accumulation due to the Warburg effect can promote Foxp3 expression by activating the GPR81 receptor or inhibiting histone deacetylases (HDACs), enhancing Treg proliferation and function [52,53]. Arginine, a critical amino acid for T cell survival and proliferation, becomes severely depleted (mainly mediated by Arg1 highly expressed on M2 macrophages and MDSCs). While this inhibits effector T cells, it relatively favors Tregs, which are more tolerant of arginine deficiency, allowing them to maintain a functional advantage [54]. In late sepsis, high expression of indoleamine 2,3-dioxygenase (IDO) drives tryptophan metabolism to generate kynurenine (Kyn). Kyn acts as an aryl hydrocarbon receptor (AhR) ligand, directly promoting Foxp3 expression and serving as a core mechanism for inducing immunosuppression and Treg expansion [55]. Free fatty acids activate Peroxisome proliferator-activated receptor gamma (PPARγ), upregulating CD36 and CPT1-mediated fatty acid oxidation (FAO), providing metabolic support for FAO-dependent Tregs [56,57]. These metabolic alterations profoundly affect the function of immune cells, including Tregs, and are closely related to sepsis severity and prognosis.

### 4.3. Shaping Forces of Immune Cell Interactions

In sepsis, the fate of Tregs, including their differentiation, expansion, survival, and immunosuppressive function, is not only modulated by soluble factors such as cytokines and metabolites, but is also directly shaped by interactions with other immune cells. During the late phase of sepsis, dendritic cells (DCs) exhibit a tolerogenic/exhausted phenotype (low co-stimulatory molecules, high PD-L1/ICOS-L, secretion of IL-10, expression of IDO1) [58]. These DCs actively induce Treg generation and provide survival signals by delivering weak antigen signals, secreting TGF-β/retinoic acid, expressing immunoregulatory ligands, and mediating IDO1-dependent tryptophan metabolism, playing a key role in maintaining the immunosuppressive microenvironment [59,60,61].

Macrophages, as key mediators in sepsis pathogenesis, exert significant regulatory effects on Tregs depending on their polarization status (M1 proinflammatory vs. M2 anti-inflammatory/reparative). In the later stages of sepsis, macrophages polarized toward the M2 phenotype actively promote Treg expansion and functional activity through multiple mechanisms, including the secretion of TGF-β and IL-10, expression of PD-L1, and arginine depletion mediated by Arg1, thereby collectively contributing to the establishment of immune paralysis [50,62,63]. Meanwhile, Tregs reciprocally secrete immunosuppressive cytokines such as IL-10, TGF-β, and IL-35; highly express CTLA-4, which facilitates trans-endocytosis of CD80/CD86 or transmits inhibitory signals; and engage in LAG-3–MHC-II interactions, all of which suppress M1 macrophage activation and drive their transition toward the M2 phenotype, forming a positive feedback loop between Tregs and M2 macrophages [64,65,66].

Furthermore, MDSCs, significantly expanded in sepsis, strongly inhibit effector T cells while relatively promoting Treg differentiation and function by producing Arg1, iNOS, ROS, TGF-β, IL-10 and expressing PD-L1 [67,68]. Overactivated neutrophils function through the formation of neutrophil extracellular traps (NETs), which induces metabolic reprogramming of naive CD4^+^ T cells through the Akt/mTOR/SREBP2 pathway to promote their conversion into Tregs [69]; specific neutrophil subsets (N2 type) may also directly exert immunosuppressive properties and cooperate with Tregs [70]. Bregs support Treg function by secreting IL-10/TGF-β/IL-35 or expressing PD-L1 [71,72].

Collectively, understanding this multi-layered regulatory network offers novel insights into therapeutic interventions for sepsis-associated immune imbalance. A comprehensive understanding of this multi-layered regulatory network offers novel insights into therapeutic interventions for sepsis-associated immune imbalance. In the early stages, therapeutic strategies may focus on inhibiting overactivated DCs and M1 macrophages or Treg function to control the cytokine storm. In later stages, careful blockade of inhibitory signaling among tolerogenic DCs, M2 macrophages, Tregs, and MDSCs (e.g., anti-PD-1/PD-L1, anti-CTLA-4) should be considered. Modulating the activity of key metabolic enzymes, including IDO1 and Arg1, also represents a promising therapeutic strategy. Future research should further delineate the specific metabolic mechanisms underlying Treg function in sepsis to establish a foundation for precise immunotherapeutic interventions.

## 5. Pathogen-Specific Treg Dynamics and Therapeutic Implications in Sepsis

The immunoregulatory role of Tregs in sepsis exhibits considerable heterogeneity. The activation and functional behavior of distinct Treg subsets are closely influenced by the type of pathogen and the anatomical site of infection. This section will explore the functional characteristics of Tregs and potential therapeutic strategies in sepsis induced by different types of pathogens (summarized in Table 1).

### 5.1. Tregs in Bacterial Sepsis

Bacterial sepsis is the most prevalent form. The most commonly isolated bacteria include *Staphylococcus aureus* (*S. aureus*), *Streptococcus pyogenes* (*S. pyogenes*), *Klebsiella* spp., *Escherichia coli* (*E. coli*), and *Pseudomonas aeruginosa* (*P. aeruginosa*) [17,73]. Ghristian Lehmann has comprehensively described the immunological mechanisms of sepsis induced by different bacterial types [74]. Broadly, bacterial pathogen-associated molecular patterns (PAMPs) bind to host cell receptors such as toll-like receptors (TLRs 4, 2, 5), triggering a robust immune response. The observed increase in Foxp3^+^ Treg proportion during early sepsis is primarily attributed to the extensive death of non-Treg cells. Recent studies also indicate that the sustained elevation of Foxp3^+^ Treg proportion in the later stages of sepsis is further influenced by ILC2-M2-mediated promotion of differentiation [50]. Tregs exert their immunomodulatory functions through multiple mechanisms. On the one hand, they secrete anti-inflammatory cytokines. For instance, IL-35 effectively suppresses Th17 cell activity, thereby mitigating LPS-induced acute liver injury [75]. Concurrently, Tregs have been found to express IL-38 (a member of the IL-1 cytokine family with anti-inflammatory properties), which promotes Th2 responses and inhibits Th1 responses to exert immunosuppressive effects. Experimental evidence demonstrates that administering recombinant human IL-38 (rmIL-38) improves survival in cecal ligation and puncture (CLP)-induced septic mice in a Treg-dependent manner [76]. On the other hand, Tregs express crucial co-inhibitory molecules on their surface, such as TIM-3, CTLA-4, and LAG-3, which are central to their immunosuppressive function. In LPS-induced acute lung injury (ALI) mice models, TIM-3^+^ Tregs upregulate IL-4 and IL-10 expression via the STAT3 signaling pathway, promoting macrophage polarization towards the M2 phenotype and subsequently alleviating lung tissue damage [77]. Clinical observations reveal that decreased TIM-3 mRNA levels in peripheral blood mononuclear cells (PBMCs) of septic patients correlate with enhanced proinflammatory cytokine release. In CLP mouse models, blocking the TIM-3 signaling pathway with anti-TIM-3 antibodies exacerbates inflammation and T cell apoptosis, worsening disease progression. Conversely, this inflammatory damage is significantly ameliorated in TIM-3-overexpressing mice [78,79]. However, the roles of these co-inhibitory molecules are complex and potentially double-edged. For example, while CTLA-4 expression on Tregs mediates immunosuppression, high levels of CTLA-4 significantly impair the clearance of Salmonella infections in mice [80]. Consistently, clinical studies further show that an elevated proportion of CTLA-4^+^ Tregs positively correlates with sepsis severity. In CLP septic mice, treatment with anti-CTLA-4 antibodies (50 µg per mouse) reduces T cell apoptosis and improves survival. Notably, however, a higher antibody dose (200 µg) worsens survival outcomes, suggesting that excessive CTLA-4 inhibition may disrupt Treg-mediated immune homeostasis and exacerbate inflammatory dysregulation [81]. Additionally, LAG-3 (CD223), primarily expressed on T cells and belonging to the immunoglobulin superfamily, is selectively expressed on Tregs. LAG-3 binds to MHC class II molecules on antigen-presenting cells (APCs), competitively inhibiting the MHC II-TCR interaction and thereby downregulating T cell-mediated immune responses [82]. It can also induce T cell cycle arrest and suppress proliferation through activation of its intracellular KIEELE motif [83]. Clinical observations indicate that LAG-3 expression on T cells often becomes more prominent during the later stages of sepsis, suggesting that blocking this pathway with LAG-3 antibodies may help reverse the immunosuppressive state characteristic of late sepsis, representing a potential therapeutic approach. TIGIT^+^Foxp3^+^ Tregs are recognized as a distinct subset of regulatory T cells with the ability to suppress Th1 and Th17 cell responses [25]. TIGIT enhances the stability of Tregs by inhibiting Akt signaling and activating FOXO1 [84]. During late-stage sepsis, TIGIT-expressing Foxp3^+^ Tregs exhibit enhanced suppressive function dependent on the IL-33/ST2/STAT6/M2 macrophage axis, suggesting that IL-33 or anti-TIGIT therapy may represent a promising strategy for the treatment of immunosuppressive sepsis [85,86]. Furthermore, as previously mentioned, CD39 enhances the immunosuppressive function of Tregs. In LPS-induced murine models of ARDS, adoptive transfer of CD39-expressing Tregs demonstrate significant protective effects, highlighting their potential as a therapeutic target [87].

Building on these pathological mechanisms, several Treg-targeted therapeutic strategies are emerging. Administration of rmIL-38 has demonstrated efficacy in improving survival in CLP-induced septic mice, acting in a Treg-dependent manner. Targeting the IL-33/ST2 axis, implicated in enhancing TIGIT^+^ Treg suppressive function during late sepsis, represents another potential strategy. Regarding co-inhibitory molecules, anti-CTLA-4 antibody treatment at a specific dose (50 µg per mouse in CLP models) reduces T cell apoptosis and improves mice survival. However, the dose-dependent effect (higher doses worsen outcomes) highlights the critical need for precise dosing to avoid disrupting immune homeostasis. Given the prominence of LAG-3 expression during late-stage sepsis immunosuppression, blocking this pathway with LAG-3 antibodies is proposed as a potential approach to reverse the immunosuppressive state. Furthermore, targeting TIGIT, potentially through anti-TIGIT therapy, is suggested as a promising strategy for treating immunosuppressive late-stage sepsis, leveraging its role in enhancing Treg stability and function. Although the clinical efficacy of immune checkpoint antibodies in sepsis treatment has not yet been fully validated, relevant antibodies have been successfully applied clinically in the field of cancer therapy. For instance, ipilimumab (an anti-CTLA-4 antibody), as an immune checkpoint blockade (ICB) drug, aimed at the deletion of Tregs, received FDA approval in 2011 [88,89]. Furthermore, some novel and potentially more effective anti-CTLA-4 antibodies, such as AGEN1181 (Phase I clinical trial, NCT03860272) and botensilimab (preclinical stage), have also demonstrated efficient targeting of Tregs, showing significant potential for antitumor therapy [90]. The effectiveness and favorable clinical safety profile demonstrated by these antibodies in cancer treatment provide an important theoretical basis and reference value for their application in sepsis immunomodulatory therapy. 

Collectively, current evidence indicates that TIM-3 primarily functions through Tregs to limit excessive inflammation during sepsis, whereas CTLA-4, LAG-3, and TIGIT are more strongly associated with the immunosuppressive state that emerges in the later stages of the disease. This phase-specific association informs the context for applying the corresponding therapeutic strategies.

### 5.2. Tregs in Viral Sepsis

Viral sepsis represents a life-threatening condition triggered by an exaggerated host response to viral infection, potentially leading to organ damage and failure [91]. Numerous viral pathogens, including influenza viruses and SARS-CoV-2, can precipitate this critical state. Pattern recognition receptors (PRRs) initiate the immune response by detecting viral components. These include TLRs, cytosolic RNA sensors (such as RIG-I and MDA5), and cytosolic DNA sensors (e.g., AIM2, IFI16, and cGAS). PRRs play a pivotal role in activating innate immunity and recruiting leukocytes upon encountering pathogens [92].

Tregs are central to maintaining immune homeostasis. Accumulating evidence suggests that during viral infections, Tregs act to constrain excessive inflammatory responses and mitigate immunopathological damage [93]. In murine models of pulmonary influenza A virus (IAV) infection, antigen-specific Tregs significantly ameliorate lung pathology and reduce immune cell infiltration post-infection [94,95,96]. IAV infection selectively promotes the accumulation and persistence of Helios-expressing Foxp3^+^ Tregs within the lungs, which exhibit potent suppressive activity against influenza virus-specific CD8^+^ T cells [97]. During IAV infection, the pulmonary CD4^+^ T cell response is primarily mediated by Th1 cells and Tregs [98]. Th1-associated responses induce the co-expression of the transcription factor T-bet in Foxp3^+^ Tregs, thereby enhancing their capacity to regulate Th1 responses within infected tissue [94]. It was further demonstrated that during IAV or lymphocytic choriomeningitis virus (LCMV) infection, Tregs respond to IFN-γ and polarize towards a Th1-like effector state. These Tregs maintain stability and effectively constrain virus-specific effector T cell function, CD8^+^ T cell proliferation, and central memory T cell formation, despite acquiring effector functions like IFNγ production [99]. Furthermore, Tregs co-expressing the T follicular helper (TFH) signature markers Bcl6 and CXCR5 have also been demonstrated to play a crucial role in enhancing the antibody response to IAV [100].

Conversely, in severe SARS-CoV-2 infection (COVID-19), intense viral replication triggers widespread CD4^+^ T cell activation and differentiation. This hyperinflammatory state critically impairs pulmonary Treg function through multi-layered mechanisms, exacerbating immunopathology [101,102]. Central to this, dysfunction is primarily driven by the COVID-19-associated cytokine storm, in which elevated IL-6 and TNF-α destabilize Tregs through epigenetic reprogramming—specifically, through STAT3-dependent deposition of repressive H3K27me3 marks at Foxp3 enhancers and synergistic suppression of maintenance signals mediated by NF-κB and PKCθ. These events collectively inhibit IL-2/STAT5-dependent Foxp3 sustenance while inducing negative regulators such as RORγt and T-bet [103,104], as evidenced by the overrepresentation of Tbet^+^ Tregs in severe COVID-19 patients [105]. Parallelly, severe pulmonary hypoxia activates hypoxia-inducible factor-1α (HIF-1α), which directly binds Foxp3 to trigger proteasomal degradation [106], thereby disrupting Foxp3 autoregulation and differentiation [106,107]. These synergistic mechanisms ultimately create a pathogenic milieu of hyperactivated CD4^+^ T cells and destabilized Tregs susceptible to functional collapse and apoptosis, fueling immunopathological cascades in severe COVID-19.

Building on these pathological mechanisms, Treg-based therapeutic strategies offer significant promise for improving outcomes in viral infections. Clinical trials demonstrate that adoptive transfer of ex vivo-expanded polyclonal iTregs effectively reduces pulmonary inflammation in patients with severe COVID-19 [108,109]. Cytokine-mediated interventions, such as IL-2 administration, represent another potential approach to induce functional Treg activation during viral infection. An ongoing clinical trial is currently assessing the efficacy of low-dose IL-2 therapy for COVID-19-associated acute respiratory distress syndrome (ARDS) [110]. Selective Treg expansion can also be achieved using anti-IL-2 monoclonal antibodies (e.g., IL-2/JES6-1 complexes), which increase Treg/Teff ratios and alleviating inflammation, offering a precise means to correct immune dysregulation in viral infections [111]. Moreover, preclinical studies with the JAK1/2 inhibitor ruxolitinib showed that it suppresses RORγt while elevating Foxp3 levels, resulting in elevated Treg counts, which suggests its therapeutic potential [110]. Furthermore, pharmacologic agents targeting key Treg-destabilizing pathways are under clinical investigation: STAT3 antagonists (e.g., WP1066; Phase II for glioblastoma [NCT05879250]) to counter IL-6-driven signaling; PKCθ inhibitors (e.g., AEB071; completed Phase I trial for transplant rejection [NCT 00545259]) to block TNF-α-mediated suppression [112]; TLR4 antagonists (e.g., E5564; a completed Phase II sepsis trial [NCT 00046072], TAK-242; a Phase III trial for sepsis-induced organ failure [NCT 00633477] showed no significant cytokine suppression in sepsis with shock/respiratory failure to attenuate viral hyperinflammation [113,114]. While these investigational agents mechanistically target Treg-stabilizing pathways, their limited clinical efficacy to date underscores the need for developing more potent therapeutic agents capable of durably maintaining Treg function in inflammatory milieus.

### 5.3. Tregs in Fungal Sepsis

Fungal pathogen infections have become an increasingly prominent issue in global public health [115]. In February 2023, the World Health Organization (WHO) released for the first time a list of priority fungal pathogens, including Cryptococcus neoformans, Candida auris, Aspergillus fumigatus, and Candida albicans [116]. In the past few decades, the incidence of pulmonary fungal infections in immunocompromised individuals has shown a significant upward trend [117].

CD4^+^ T cell subsets play a pivotal role in the immune response to fungal infections. Protective immunity is predominantly mediated by Th1 and Th17 cells. Specifically, IFN-γ produced by Th1 cells activates classical macrophage pathways and facilitates neutrophil recruitment, thereby promoting pathogen clearance. Meanwhile, IL-17 secreted by Th17 cells enhances mucosal immune responses, contributing synergistically to host defense against fungal pathogens [118]. In contrast, Th2 cells have detrimental effects by secreting IL-4, IL-5, and IL-13, which promote eosinophilia, induce M2 macrophage polarization, and exacerbate allergic inflammation [119]. Consequently, Th2-type responses not only inhibit effective fungal clearance but also directly contribute to immunopathological damage associated with uncontrolled pulmonary fungal infections [120].

Within this complex regulatory network, Tregs exhibit significant complexity and dual regulatory effects. Numerous studies indicate that Tregs can exert protective functions by suppressing detrimental Th2 responses or promoting Th1/Th17-mediated inflammation. For example, in a Cryptococcus neoformans infection model, conditional depletion of Foxp3^+^ Tregs resulted in elevated pulmonary Th2 cell counts, indicating that Tregs restrict the expansion of Th2 cells induced by cryptococcal infection [121,122]. Further studies have demonstrated that Tregs are recruited to infected lung tissues through CCR5- and IRF4-dependent pathways, thereby suppressing pathogenic Th2-mediated immune responses [121,123]. Supporting this, enhancing Tregs via IL-2/anti-IL-2 antibody complexes or disrupting Treg-mediated suppression of Th2 cells accelerates disease progression and mortality in invasive cryptococcosis models [124]. However, the role of Tregs is also context-dependent. In an oropharyngeal Candida albicans infection model, Tregs effectively suppressed mucosal fungal infection by promoting Th17 secretion of IL-17A/F and IL-22 [125,126]. In gastric C. albicans infection, Treg-mediated regulation exhibits temporal dynamics: during early infection, nTregs control excessive inflammation by modulating polymorphonuclear neutrophil (PMN) activity, thereby limiting initial fungal growth and local inflammation. However, as infection progresses, antigen-specific iTregs are induced and suppress Th17-mediated protective immunity [32].

Notably, multiple fungal pathogens subvert host immunity by exploiting Treg pathways. Candida albicans induces IL-10 production and promotes Treg survival through TLR2-PI3K-Foxp3 signaling, thereby impairing infection clearance [127]. Similarly, both Cryptococcus species leverage TLR2 activation to expand Tregs: C. gattii infection models show TLR2 agonism drives Treg differentiation while suppressing Th1/Th17 responses [128], whereas C. neoformans directly promotes Treg expansion via GXM/melanin-dependent TLR2 signaling and reinforces differentiation through CPL1-induced M2 macrophage polarization [129]. Aspergillus fumigatus similarly exploits TLR2 to induce Treg expansion [130], with its α-(1,3)-glucan further enhancing Treg expansion by triggering PD-L1 on dendritic cells; consequently, PD-1/PD-L1 blockade restores antifungal immunity [131].

Beyond Treg manipulation, impaired Notch signaling in C. neoformans models exacerbates fungal burdens by selectively compromising Th1/Th2 (but not Th17) responses [132], underscoring the non-redundant role of Th1 immunity [133]. Additionally, IL-27 demonstrates dual immunomodulatory functions—promoting Th1 differentiation while suppressing Th17 and enhancing Treg activity—suggesting therapeutic potential in fungal control [134].

Over the past decades, despite significant advances in understanding of host immune mechanisms against pathogenic infections, research into fungal infection and novel antifungal drug development has progressed slowly. While immunotherapies exert advantages such as improved safety and lower resistance risk, the field of fungal immunotherapy remains nascent. Current insights into CD4^+^ T cell subset regulation during fungal infections reveal three translationally promising strategies: (1) enhancing protective immunity—IFN-γ therapy to augment Th1-mediated responses has significantly improved clinical outcomes in fungal sepsis. In the context of renal transplant patients, IFN-γ should be administrated early during invasive aspergillosis infection [135,136,137]. (2) Blocking pathogen immune evasion—developing interventions like TLR2 pathway antagonists to inhibit pathogen-induced Treg expansion and IL-10 production, thereby restoring Th1/Th17 cell antifungal efficacy. Notably, OPN-305, a humanized anti-TLR2 IgG4 monoclonal antibody, has advanced in Phase II trials (NCT01794663) for the prevention of delayed renal graft function, which suggests a potential clinical benefit in fungal sepsis. (3) Precise immunobalance modulation—coordinating Th1 differentiation and Treg activity using pleiotropic factors (IFN-γ, TNF-α, or IL-27 or modulating Notch signaling to enhance Th1 responses. This approach aims to optimize pathogen clearance while controlling excessive inflammation, offering dynamic interventions for complex infections.

## 6. The Phase-Dependent Strategy of Treg in Immune Regulation of Sepsis

Given the dual-phase role of Tregs during the septic pathological process, this necessitates precise identification of the immune status phenotype to determine the optimal therapeutic window for Treg intervention. The pronounced immunopathological heterogeneity in sepsis manifests as dynamic alternation or coexistence of hyperinflammation and immunoparalysis, requiring stratification through multidimensional biomarkers.

The hyperinflammatory phase can be identified by an elevated IFN-γ/IL-10 ratio or significantly increased ferritin levels indicating macrophage activation syndrome (MAS) [138,139]. Early Treg intervention during this phase can help mitigate cytokine storms. An IL-6 level exceeding 1000 pg/mL within 24 h of admission serves as a hallmark of the early hyperinflammatory phase, where Tregs can exert organ-protective effects. Crucially, a rapid decline in IL-6 levels signals impending immunosuppression risk and necessitates therapeutic reassessment [140,141].

Conversely, the immunoparalytic phase, marked by monocyte human leukocyte antigen (HLA)-DR expression lower than 8000 molecules per cell for 2 days, as revealed by specific monoclonal antibodies, suggests that Treg activity may amplify immunosuppressive effects [142]. In addition, gene expression endotyping has identified two distinct sepsis response signatures: SRS2, characterized by immunocompetence, and SRS1, characterized by immunosuppression. In this cohort, glucocorticoid treatment, an immunosuppressive therapeutic intervention, was significantly associated with a 7.9-fold increased risk of mortality (OR = 7.9) among patients exhibiting the SRS1 subtype, strongly cautioning against Treg-based interventions during immunoparalysis [143]. An absolute lymphocyte count ≤ 1.1 × 10^3^/μL provides direct evidence of immunoparalysis, wherein Tregs may potentiate immune exhaustion [144,145].

Synergistic monitoring of PCT (procalcitonin) and CRP (C-reactive protein) enhances phenotypic discrimination: persistently elevated PCT coupled with CRP > 150 mg/L indicates sustained inflammation (a potential window for Treg intervention), whereas declining PCT concomitant with lymphocytopenia signifies the immunoparalytic phase, mandating a therapeutic shift towards immune-stimulating strategies that counteract Treg-mediated suppression (NCT04920565, NCT04013269) [146].

## 7. Summary and Future Perspectives

In conclusion, Tregs represent central components within the immune dysregulation network observed in sepsis. Their functional characteristics exhibit substantial dynamism and heterogeneity across temporal phases, anatomical locations, and pathogen-specific contexts. These features necessitate a precise, individualized, and time-dependent approach when targeting Treg cells therapeutically, aiming to reestablish immune homeostasis and enhance clinical outcomes for sepsis patients. Critically, clinical translation efforts are bolstered by over 15 years of accumulated evidence: more than 25 early-stage trials (primarily in autoimmunity, transplantation, and graft-versus-host disease) have conclusively demonstrated the feasibility, safety, and long-term tolerability of Treg-based therapies [147]. Building on this foundation, future progress requires deeper characterization of sepsis-specific Treg heterogeneity, development of pathogen-tailored immunotherapies guided by real-time immune monitoring, and innovative approaches to selectively harness Tregs’ protective functions while mitigating detrimental immunosuppression. The emergence of commercially developed strategies—exemplified by Cellenkos’ use of unmodified umbilical cord blood-derived Tregs (CK0801) to treat COVID-19-associated acute respiratory distress syndrome (ARDS)—further validates the therapeutic potential of Treg modulation and highlights its adaptability to sepsis-related organ injury [147,148]. Collectively, these advances provide a robust framework for accelerating targeted interventions against this devastating syndrome.

## Figures and Tables

**Figure 1 ijms-26-07436-f001:**
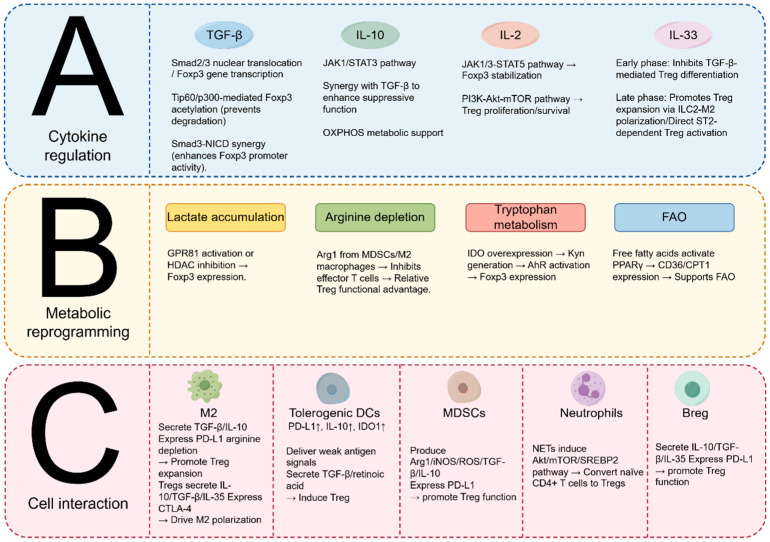
(**A**–**C**) The immunoregulatory network that governs Treg differentiation and maintenance in sepsis cytokine signaling pathways, including TGF-β/Smad3, IL-10/STAT3, IL-2/STAT5, and IL-33/ST2, critically drive Treg differentiation and functional stability. These cytokines originate from multiple sources and act in concert to modulate Foxp3 expression and Treg expansion. Interactions with immune cells such as tolerogenic dendritic cells (DCs), M2-polarized macrophages, myeloid-derived suppressor cells (MDSCs), and regulatory neutrophils shape Treg activity. DCs utilize PD-L1 and IDO to promote immune tolerance, macrophages secrete TGF-β and IL-10 while depleting arginine via Arg1, and MDSCs further contribute immunosuppressive cytokines and metabolic modulation to support Tregs. Metabolic reprogramming enables Tregs to sustain their suppressive function within the inflammatory milieu of sepsis. This includes utilization of lactate, enhanced oxidative phosphorylation (OXPHOS), arginine depletion, kynurenine signaling via the aryl hydrocarbon receptor (AhR), and fatty acid oxidation.

**Table 1 ijms-26-07436-t001:** Tregs in Sepsis: Properties, Functions, and Therapeutic Strategies.

Sepsis Type	Treg Biological Properties & Functions	Potential Therapeutic Strategies
**Bacterial**	**Early phase**: Foxp3^+^ Treg proportion ↑ (due to non-Treg cell death) **Late phase**: Sustained Foxp3^+^ Tregs elevation via ILC2-M2 axis **Functions**: - IL-35 reduces liver injury - IL-38 improves survival - TIM-3^+^Tregs reduce lung injury - CTLA-4^+^Tregs, LAG-3^+^Tregs and TIGIT^+^Tregs impair bacterial clearance, increase sepsis severity	1. **IL-38 therapy**: Recombinant IL-38 (Treg-dependent survival improvement 2. **Co-inhibitory targeting**: - Anti-CTLA-4: Low dose (50 µg/mouse) ↓ T-cell apoptosis - Anti-LAG-3: Reverses late-phase immunosuppression - Anti-TIGIT/IL-33: Enhances Treg function 3. **Phase-specific**: - Early: Enhance TIM-3 - Late: Block CTLA-4/LAG-3
**Viral**	**Protective roles in Influenza**: Suppresses hyperinflammation: - Lung Helios^+^Tregs inhibit CD8^+^ T-cell responses - Bcl6^+^CXCR5^+^ Tregs enhances antibody responses - Th1-like polarization (T-bet^+^) Tregs inhibit CD8^+^ T-cell responses **Pathology in COVID-19**: - Impaired Treg function (Foxp3 degradation via HIF-1α, IL-6/STAT3, TGF-β/NF-Κb & PKCθ)	1. **Adoptive Treg transfer**: iTregs reduce lung inflammation (COVID-19) 2. **Cytokine therapy**: - Low-dose IL-2: Treg proliferation↑ - Anti-IL-2 mAbs: Selective Treg expansion 3. **Signaling modulation**: - JAK1/2 inhibitors (e.g., ruxolitinib: FOXP3 ↑/RORγt ↓) 4. **Foxp3 stabilization**: Inhibit HIF-1α or STAT3 antagonists, PKCθ inhibitors, TLR4 antagonists
**Fungal**	**Protective**: - Suppresses Th2 (*C. neoformans*) - Promotes Th17 IL-17/IL-22 (*C. albicans*) **Pathogenic**: - Pathogen-induced Treg expansion via TLR2/IL-10 (*C. albicans*, *C. gattii*) - α-(1,3)-glucan enhance Treg expansion by triggering PD-L1 (*Aspergilus fumigatus*) - Late-phase iTregs suppress Th17 (gastric candidiasis)	1. **Boost Th1**: IFN-γ therapy (candidiasis/aspergillosis) 2. **Block evasion**: TLR2 antagonists iTreg/IL-10 ↓ 3. **Immunomodulation**: - IL-27 balances Th1/Treg/Th17 - Notch modulation Th1 ↑ 4. **Phase-targeted**: - Early: Enhance nTregs - Late: Inhibit iTregs

## Data Availability

No new data were created or analyzed in this study.

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
