# Peer review of "The Multifaceted Role of Regulatory T Cells in Sepsis: Mechanisms, Heterogeneity, and Pathogen-Tailored Therapies"

_ijms, 2025, doi:10.3390/ijms26157436_

Round 1
Reviewer 1 Report
Comments and Suggestions for Authors
This review article, "The Multifaceted Role of Regulatory T Cells in Sepsis: Mechanisms, Heterogeneity, and Pathogen-Tailored Therapies," offers an excellent and timely overview of the current understanding of regulatory T cells (Tregs) in sepsis. The authors have undertaken a challenging task of disentangling the biphasic immune dysregulation in sepsis and have positioned Tregs at the center of both its hyperinflammatory and immunosuppressive phases, achieving this with notable clarity and depth. The review is highly informative, grounded in strong mechanistic knowledge, and structured thoughtfully to benefit both immunologists and clinicians.
The sections focused on Treg heterogeneity, differentiation states, and effector specialization are particularly well-developed, reflecting a strong command over the molecular pathways that shape Treg phenotype and function. The detailed exploration of cytokine signaling and the profiles of transcription factors (including T-bet, IRF4, STAT3, GATA3, and Helios), alongside metabolic reprogramming factors (like lactate accumulation and arginine depletion), provides valuable insights to the field. Additionally, the discussion surrounding immune cell crosstalk, including interactions of Tregs with dendritic cells, macrophages, MDSCs, and neutrophils, effectively highlights the complex regulation of Tregs in the context of sepsis. These insights are well-supported by existing literature and adequately referenced.
The sections addressing bacterial, viral, and fungal sepsis are particularly insightful, displaying a careful integration of mechanistic immunology with pathogen-specific host responses. The manuscript also thoughtfully incorporates therapeutic implications, from cytokine-based interventions to immune checkpoint modulation. Of particular note is the section on viral sepsis, which discusses ongoing clinical research, including adoptive Treg transfer in COVID-19, low-dose IL-2 therapy, and anti-IL-2 monoclonal antibody approaches. These examples contribute significantly to the translational relevance of the review.
However, while the manuscript effectively integrates clinical trial data in the context of viral sepsis, particularly regarding COVID-19, similar translational perspectives are lacking in the sections on bacterial and fungal sepsis. Including available clinical observations, therapeutic studies, or trials focused on Treg modulation in these contexts would enhance the overall balance of the review and strengthen its applicability to diverse sepsis etiologies. Even where trials are limited, brief commentary on clinical challenges or translational barriers would enrich these sections.
Furthermore, the discussion of Treg instability and plasticity, particularly under the inflammatory and hypoxic conditions characteristic of sepsis, could be expanded. Although the viral sepsis section briefly mentions Treg destabilization through HIF-1α and Furin deficiency, the broader concept of Treg plasticity, such as the loss of Foxp3, conversion into effector-like cells, or the acquisition of pro-inflammatory features, merits further exploration. Given the increasing evidence of such instability in inflammatory disorders, a brief commentary on its implications for Treg-based therapies in sepsis would be beneficial. It would add depth to the already strong therapeutic sections.
The manuscript emphasizes the need for precisely timed and pathogen-specific modulation of Treg cells. However, it currently offers no guidance on how such timing might be determined in practice. A discussion of candidate biomarkers, such as cytokine profiles and transcriptomic signatures, could provide readers with insights into how dynamic immune monitoring might inform the timing of Treg-directed therapies in clinical practice. This would further strengthen the manuscript's translational message.
The fungal sepsis section is innovative and presents promising conceptual frameworks for immunotherapy. I believe the authors could add several connections between specific fungal immune evasion mechanisms (such as TLR2-mediated IL-10 induction by Candida albicans) and the proposed therapeutic strategies. Additionally, while the review suggests approaches like IFN-γ therapy and Notch signaling modulation, it does not clarify whether these strategies have been investigated in fungal sepsis models. Adding relevant preclinical data or explaining the status of the theoretical strategy would enhance the transparency and impact of this significant research.
In conclusion, this is an excellently well-written and thoroughly researched review that makes a meaningful contribution to the fields of immunology and critical care. The authors deserve recognition for their scientific rigor and clarity in presenting this complex topic. Addressing the highlighted points, particularly regarding broader clinical insights across all sepsis types, elaborating on Treg instability, and providing guidance on therapeutic timing, would further enhance the work's overall contribution.
I would like to extend my gratitude to the authors for their timely and insightful review.
Author Response
For review article
|
Response to Reviewer 1 Comments
|
||||
|
1. Summary |
|
|
||
|
Thank you very much for taking the time to review this manuscript! Please find the detailed responses below and the corresponding revisions/corrections highlighted/in track changes in the re-submitted files.
|
||||
|
2. Point-by-point response to Comments and Suggestions for Authors |
|
|
||
|
|
|
|
||
“Current insights into CD4⁺ T cell subset regulation during fungal infections reveal three translationally promising strategies: 1) Enhancing protective immunity: IFN-γ therapy to augment Th1-mediated responses has significantly improved clinical outcomes in fungal sepsis [135-137]. 2) Blocking pathogen immune evasion: Developing interventions like TLR2 pathway antagonists to inhibit pathogen-induced Treg expansion and IL-10 production, thereby restoring Th1/Th17 cell antifungal efficacy. Notably, OPN-305, a humanized anti-TLR2 IgG4 monoclonal antibody has advances in Phase II trials (NCT01794663) for the prevention of delayed renal graft function, which suggests a potential clinical benefit in fungal sepsis.”
Comments 2: Furthermore, the discussion of Treg instability and plasticity, particularly under the inflammatory and hypoxic conditions characteristic of sepsis, could be expanded. Although the viral sepsis section briefly mentions Treg destabilization through HIF-1α and Furin deficiency, the broader concept of Treg plasticity, such as the loss of Foxp3, conversion into effector-like cells, or the acquisition of pro-inflammatory features, merits further exploration. Given the increasing evidence of such instability in inflammatory disorders, a brief commentary on its implications for Treg-based therapies in sepsis would be beneficial. It would add depth to the already strong therapeutic sections.
Response 2: We sincerely appreciate the reviewer for this insightful suggestion. In response, we have significantly expanded the discussion on Treg instability and plasticity within the context of sepsis, with a focused analysis on COVID-19-induced viral sepsis as originally presented in Section. Please see the changes detailed below: Targeted Exploration of Treg Destabilization Mechanisms (line 372): Central to this dysfunction is primarily driven by the COVID-19-associated cytokine storm, in which elevated IL-6 and TNF-α destabilize Tregs through epigenetic reprogramming—specifically, through STAT3-dependent deposition of repressive H3K27me3 marks at Foxp3 enhancers and synergistic suppression of maintenance signals mediated by NF-κB and PKCθ. These events collectively inhibit IL-2/STAT5-dependent Foxp3 sustenance while inducing negative regulators such as RORγt and T-bet [103, 104], as evidenced by the the overrepresentation of Tbet+ Tregs in severe COVID-19 patients [105]. Parallelly, severe pulmonary hypoxia activates hypoxia-inducible factor-1α (HIF-1α), which directly binds Foxp3 to trigger proteasomal degradation [106], thereby disrupting Foxp3 autoregulation and differentiation [106, 107]. These synergistic mechanisms ultimately create a pathogenic milieu of hyperactivated CD4⁺ T cells and destabilized Tregs susceptible to functional collapse and apoptosis, fueling immunopathological cascades in severe COVID-19. Rationale for Removal of Furin Discussion: We appreciate the reviewer’s mention of Furin. Upon careful reconsideration, we have removed the specific discussion on Furin deficiency from the manuscript. This decision was based on Furin’s dual and opposing roles in COVID-19 pathogenesis: Proviral Role: Furin is essential for SARS-CoV-2 spike protein processing and viral cellular entry. Treg-Stabilizing Role: Furin acts as a proprotein convertase crucial for Foxp3 stability. Given this inherent therapeutic conflict, targeting Furin to stabilize Tregs in COVID-19 sepsis is clinically untenable. Therefore, we are afraid that its inclusion could be misleading regarding therapeutic feasibility. “T cell-specific deletion of Furin, a proprotein convertase essential for Foxp3 stability, results in impaired Foxp3 and Tbx21 expression, promoting Treg differentiation into other T cell subsets [92].”(line 369) Therapeutic Implications Section (line 398): “Furthermore, pharmacologic agents targeting key Treg-destabilizing pathways are under clinical investigation: STAT3 antagonists (e.g., WP1066; Phase II for glioblastoma [NCT05879250]) to counter IL-6-driven signaling; PKCθ inhibitors (e.g., AEB071; completed Phase I trial for transplant rejection [NCT 00545259]) to block TNF-α-mediated suppression [112]; TLR4 antagonists (e.g., E5564; competed Phase II sepsis trial [NCT 00046072], TAK-242; Phase III trial for sepsis-induced organ failure [NCT 00633477] showed no significant cytokine suppression in sepsis with shock/respiratory failure] to attenuate viral hyperinflammation [113, 114]. While these investigational agents mechanistically target Treg-stabilizing pathways, their limited clinical efficacy to date underscores the need for developing more potent therapeutic agents capable of durably maintaining Treg function in inflammatory milieus.” Comments 3: The manuscript emphasizes the need for precisely timed and pathogen-specific modulation of Treg cells. However, it currently offers no guidance on how such timing might be determined in practice. A discussion of candidate biomarkers, such as cytokine profiles and transcriptomic signatures, could provide readers with insights into how dynamic immune monitoring might inform the timing of Treg-directed therapies in clinical practice. This would further strengthen the manuscript's translational message. Response 3: We sincerely appreciate the reviewer’s expert comments. We fully agree with this point. The detail information has been added. Please see the changes blew (line 483): “6. The phase-dependent strategy of Treg in immune regulation of sepsis Given the dual-phase role of Tregs during the septic pathological process, this necessitates precise identification of the immune status phenotype to determine the optimal therapeutic window for Treg intervention. The pronounced immunopathological heterogeneity in sepsis manifests as dynamic alternation or coexistence of hyperinflammation and immunoparalysis, requiring stratification through multidimensional biomarkers. The hyperinflammatory phase can be identified by an elevated IFN-γ/IL-10 ratio or significantly increased ferritin levels indicating macrophage activation syndrome (MAS) [138, 139]. Early Treg intervention during this phase can help mitigate cytokine storms. An IL-6 level exceeding 1000 pg/mL within 24 hours of admission serves as a hallmark of the early hyperinflammatory phase, where Tregs can exert organ-protective effects. Crucially, a rapid decline in IL-6 levels signals impending immunosuppression risk and necessitates therapeutic reassessment [140, 141]. Conversely, the immunoparalytic phase, marked by monocyte human leukocyte antigen (HLA)-DR expression <8,000 monoclonal antibodies (mAb) per cell for 2 d, suggests that Treg activity may amplify immunosuppressive effects [142]. In addition, gene expression endotyping has identified two distinct sepsis response signatures: SRS2, characterized by immunocompetence, and SRS1, characterized by immunosuppression. In this cohort, glucocorticoid treatment, an immunosuppressive therapeutic intervention, was significantly associated with a 7.9-fold increased risk of mortality (OR = 7.9) among patients exhibiting the SRS1 subtype, strongly cautioning against Treg-based interventions during immunoparalysis [143]. An absolute lymphocyte count ≤ 1.1 × 10³/μL provides direct evidence of immunoparalysis, wherein Tregs may potentiate immune exhaustion [144, 145]. Synergistic monitoring of PCT (procalcitonin) and CRP (C-reactive protein) enhances phenotypic discrimination: Persistently elevated PCT coupled with CRP > 150 mg/L indicates sustained inflammation (a potential window for Treg intervention), whereas declining PCT concomitant with lymphocytopenia signifies the immunoparalytic phase, mandating a therapeutic shift towards immune-stimulating strategies that counteract Treg-mediated suppression (NCT04920565, NCT04013269) [146].” |
||||
|
|
||||
|
||||
Reviewer 2 Report
Comments and Suggestions for Authors
The article analyzes the dual role of T-regulatory cells in the development of sepsis.
It is known that Regulatory T cells (Tregs) play a complex and sometimes contradictory role in sepsis. While Tregs are crucial for maintaining immune homeostasis and preventing excessive inflammation, their increased presence and activity in sepsis can contribute to immune suppression, leading to impaired pathogen clearance and increased susceptibility to secondary infections.
The abstract reflects the content of the review. The keywords are adequate. It is recommended at the discretion of the authors to indicate the depth of the information search and indicate the databases that were used to search and select articles (Pubmed, Scopus). In addition, you need to explain what is special and unique about your review, given the fact that we can find many similar reviews.
- Introduction.
The information on the prevalence of sepsis and its huge contribution to global mortality is presented. General information on Tregs is analyzed, as well as the reversible effect of Tregs in the dynamics of sepsis development. At the end of this section, it is recommended to clearly formulate the purpose of the review, as well as this fundamental difference from other similar reviews.
It is also important to emphasize the species affiliation of the markers analyzed in this article – whether this concerns humans or whether animal models were also analyzed.
- Treg Heterogeneity.
This section analyzes information about the subpopulations of Tregs, as well as the main generally accepted surface and intracellular markers of these cells.
It probably makes sense to add information about TIGIT+Tregs, a highly suppressive subset of Tregs, are known to expand during the late phase of sepsis.
- Immunoregulatory Network of Tregs in Sepsis.
This section analyzes information about the contribution of Tregs to the development of sepsis. First of all, data on the role of cytokines in the regulation of Tregs differentiation are presented (3.1. Core Driving Role of Cytokines). The contribution of TGF-β1, IL-10, IL-2, IL-33 – the main Tregs-associated cytokines – is analyzed.
Next, information is provided on the metabolic processes that regulate the development of Tregs (3.2. Critical Impact of Metabolic Reprogramming). In particular, the role of arginine, lactate, tryptophan, and free fatty acids is considered. The next item is an analysis of 3.3. Shaping Forces of Immune Cell Interactions. It is obvious that Tregs interact with other cells of the immune system, so the authors analyze the contribution of dendritic cells, macrophages, and MDSC.
- Pathogen-Specific Treg Dynamics and Therapeutic Implications in Sepsis
This section analyzes the role of Tregs in bacterial sepsis, viral sepsis, and Fungal Sepsis. According to the reviewer, there is a lack of information on the role of Tregs in the development of prion disease, as well as on the role of Tregs in the development of sterile sepsis (SIRS).
Summary and Future Perspectives
This section should be supplemented, perhaps by presenting existing patents on the use of cell therapy in sepsis therapy, or successful cases of sepsis therapy using Treg modulation.
The review contains 116 references.
Explain the principle by which you selected the sources if a superficial search showed the following articles on the topic of the review, but they are not cited by you.
Gao X, Cai S, Li X, Wu G. Sepsis-induced immunosuppression: mechanisms, biomarkers and immunotherapy. Front Immunol. 2025 Apr 29;16:1577105. doi: 10.3389/fimmu.2025.1577105.
Liu D, Huang SY, Sun JH, Zhang HC, Cai QL, Gao C, Li L, Cao J, Xu F, Zhou Y, Guan CX, Jin SW, Deng J, Fang XM, Jiang JX, Zeng L. Sepsis-induced immunosuppression: mechanisms, diagnosis and current treatment options. Mil Med Res. 2022 Oct 9;9(1):56. doi: 10.1186/s40779-022-00422-y. PMID: 36209190; PMCID: PMC9547753.
Jiang LN, Yao YM, Sheng ZY. The role of regulatory T cells in the pathogenesis of sepsis and its clinical implication. J Interferon Cytokine Res. 2012 Aug;32(8):341-9. doi: 10.1089/jir.2011.0080.
Kessel A, Bamberger E, Masalha M, Toubi E. The role of T regulatory cells in human sepsis. J Autoimmun. 2009 May-Jun;32(3-4):211-5. doi: 10.1016/j.jaut.2009.02.014.
Your review will be more readable if you include drawings in the text or a graphic abstract.
Author Response
|
1. Summary |
|
|
||
|
Thank you very much for taking the time to review this manuscript! Please find the detailed responses below and the corresponding revisions/corrections highlighted/in track changes in the re-submitted files.
|
||||
|
2. Point-by-point response to Comments and Suggestions for Authors |
|
|
||
|
|
|
|
||
Comments 2: 1. Introduction. The information on the prevalence of sepsis and its huge contribution to global mortality is presented. General information on Tregs is analyzed, as well as the reversible effect of Tregs in the dynamics of sepsis development. At the end of this section, it is recommended to clearly formulate the purpose of the review, as well as this fundamental difference from other similar reviews. Response 2: We sincerely appreciate the reviewer’s expert comments. We fully agree with this point (line 58). Therefore, this review aims to critically evaluate the pathogen-specific modulation of Treg heterogeneity and its functional consequences for immune dysregulation across the sepsis continuum. Crucially, we will emphasize how deciphering this pathogen-driven heterogeneity is pivotal for developing targeted, etiology-tailored immunomodulatory strategies. Comments 3: It is also important to emphasize the species affiliation of the markers analyzed in this article – whether this concerns humans or whether animal models were also analyzed. Response 3: We sincerely appreciate the reviewer’s expert comments. We fully agree with this point. The detail information has been added (line 90). Compared with cTregs, eTregs exhibit enhanced functional adaptability and phenotypic diversity, characterized by the upregulation of specific transcription factors, chemokine receptors, and effector molecules, which enables effective suppression of immune responses within distinct microenvironments [4]. For example, during Th1-type inflammation, the transcription factor T-bet—indispensable for Th1 differentiation—is also induced in Tregs in both humans and mice and plays a pivotal role in suppressing Th1-mediated immune responses [5, 10]. Similarly, interferon regulatory factor 4 (IRF4) expression in Tregs is essential across species for the effective inhibition of Th2-type immune responses [15, 16]. In addition, specific deletion of STAT3 in Tregs in mice results in impaired regulation of Th17-associated immune reactions, highlighting its critical functional role in this model (with analogous pathways implicated in humans) [17, 18]. Moreover, although the transcription factor Bcl6 is a central regulator of follicular helper T cell (TFH) development, it is also expressed in Foxp3+ Tregs in both humans and mice, giving rise to a distinct subset known as follicular regulatory T cells (TFRs), which predominantly mediate immunosuppression during Th2-type immune responses [19]. GATA3 is a transcription factor driving type 2 immunity. GATA3 expression in Treg cells in both humans and mice is especially important for their tempering of Th2 responses in the intestine and the skin [20-22] Furthermore, Helios serves as a key signature marker of mouse and human tTregs, associated with a pronounced effector-like phenotype, enhanced suppressive capacity [23], and essential stability for Treg suppressive function [6,24].
|
||||
|
|
||||
|
||||
|
Comments 5: 3. Pathogen-Specific Treg Dynamics and Therapeutic Implications in Sepsis. This section analyzes the role of Tregs in bacterial sepsis, viral sepsis, and Fungal Sepsis. According to the reviewer, there is a lack of information on the role of Tregs in the development of prion disease, as well as on the role of Tregs in the development of sterile sepsis (SIRS). |
||||
|
|
||||
|
Response 5: We sincerely appreciate the reviewer's insightful comments regarding Tregs in prion diseases and sterile sepsis. To better contextualize our pathogen-focused scope, we provide the following explanatory notes based on established definitions: We recognize prion diseases (e.g., Creutzfeldt-Jakob disease) as protein-misfolding neurodegenerative disorders characterized by pathological prion protein accumulation. Their primary pathology centers on neuronal apoptosis and spongiform degeneration rather than systemic inflammatory responses to infection or tissue injury. While the immunobiology of prion diseases represents a fascinating area of study (including potential Treg involvement), the pathogenic mechanisms differ fundamentally from infection-driven sepsis processes covered in our review. We therefore focused our analysis on sepsis as defined by current consensus criteria We acknowledge the significance of sterile inflammatory conditions. However, our Section 4 titled” Section 4 ("Pathogen-Specific Treg Dynamics and Therapeutic Implications in Sepsis") specifically examines Treg dynamics in infection-driven sepsis triggered by pathogen-associated molecular patterns (PAMPs). However Sterile sepsis, driven by damage-associated molecular patterns (DAMPs) from non-infectious tissue injury (e.g., severe pancreatitis, major trauma), falls outside this pathogen-centric framework. Furthermore, the Sepsis-3 consensus definition (Singer et al., JAMA 2016; 315:801-810) mandates that sepsis arises from "life-threatening organ dysfunction caused by a dysregulated host response to infection." Therefore, we agree sterile sepsis are critically important, but they represent a distinct pathophysiological entity best classified as "sepsis-like syndrome". We fully acknowledge the significance of the reviewer's points. While beyond the current scope of our pathogen-focused review, we agree that comparative studies investigating Treg biology across diverse inflammatory contexts, including classical infection-driven sepsis (PAMP-triggered) versus sterile sepsis-like syndromes (DAMP-triggered) represent a vital future research avenue. Understanding how Treg function differs or converges in these settings could yield novel immunomodulatory insights.
|
||||
Round 2
Reviewer 2 Report
Comments and Suggestions for Authors
The authors presented a revised version of the article and answered questions and comments.
In particular, a new section, Literature Search Methodology, was added. After the introduction, the purpose of the review was clearly formulated. Information on the species affiliation of Treg markers was added, and information on the TIGIT+Treg marker was added.
The authors convincingly answered questions regarding the role of Tregs in the development of prion diseases and sterile sepsis.
The authors also supplemented the abstract with information on successful cases of Treg-targeted therapy.
The authors answered questions about the links and explained the method for selecting articles for citation. In addition, a figure was added to the review, summarizing current understanding of the Immunoregulatory network regulating Treg differentiation and maintenance in sepsis.
In general, this version of the review can be published.
Author Response
We sincerely appreciate the reviewer's expert comments!